# Reasons for readmission after hospital discharge in patients with chronic diseases— Information from an international dataset

**Hans-Peter Brunner-La Rocca**[1]*, **Carol J. Peden**[2], **John Soong**[3], **Per Arne Holman**[4], **Maria Bogdanovskaya**[5], **Lorna Barclay**[5]

**1** Department of Cardiology, Maastricht University Medical Center, Maastricht, The Netherlands, **2** Center for Health System Innovation, Keck Medicine of USC, Los Angeles, California, United States of America, **3** NIHR CLAHRC for Northwest London Team, Imperial College London, Chelsea and Westminster Hospital NHS Foundation Trust, London, United Kingdom, **4** Department of Patients safety and Research, Lovisenberg Diaconal Hospital, Oslo, Norway, **5** Dr Foster Telstra Health, London, United Kingdom

* hp.brunnerlarocca@mumc.nl

## Abstract

### Background

Chronic diseases are increasingly prevalent in Western countries. Once hospitalised, the chance for another hospitalisation increases sharply with large impact on well-being of patients and costs. The pattern of readmissions is very complex, but poorly understood for multiple chronic diseases.

### Methods

This cohort study of administrative discharge data between 2009–2014 from 21 tertiary hospitals (eight USA, five UK, four Australia, four continental Europe) investigated rates and reasons of readmissions to the same hospital within 30 days after unplanned admission with one of the following chronic conditions; heart failure; atrial fibrillation; myocardial infarction; hypertension; stroke; chronic obstructive pulmonary disease (COPD); bacterial pneumonia; diabetes mellitus; chronic renal disease; anaemia; arthritis and other cardiovascular disease. Proportions of readmissions with similar versus different diseases were analysed.

### Results

Of 4,901,584 admissions, 866,502 (17.7%) were due to the 12 chronic conditions. In-hospital, 43,573 (5.0%) patients died, leaving 822,929 for readmission analysis. Of those, 87,452 (10.6%) had an emergency 30-day readmission, rates ranged from 2.8% for arthritis to 18.4% for COPD. One third were readmitted with the same condition, ranging from 53% for anaemia to 11% for arthritis. Reasons for readmission were due to another chronic condition in 10% to 35% of the cases, leaving 30% to 70% due to reasons other than the original 12 conditions (most commonly, treatment related complications and infections). The chance of being readmitted with the same cause was lower in the USA, for female patients, with increasing age, more co-morbidities, during study period and with longer initial length of stay.

**Data Availability Statement:** Data cannot be made publicly available as the terms and conditions of this collaboration restricted the availability of

original data as follows: 1) Only certain categories of people are permitted to access and use the "Analytical Tool" (that is, the software processing input data and generating output data for providing benchmarking to each hospital for quality improvements). These are defined as: "Permitted Users" means the Participant's directors, employees and independent contractors who are authorised by the Participant to access and use the Analytical Tool for the purposes provided in this Agreement. 2) Access to and use of the Analytical Tool itself is granted only for the "Term" and for specific purposes, being "accessing and viewing the Output Data and to use the Output Data for the purpose of participating in and reviewing the results of the Project". The restriction is related to the contract between the participating hospitals and Dr. Foster Telstra Health. There it was specified who has access to the data and that they may not be shared with other people as described above. Other researchers can request access to the data as long as they fulfil the requirements of the contract. Contracting the responsible person in each individual hospital is possible, but the authors cannot guarantee that each hospital will provide the data. This is the responsibility of each hospital. It would be all administrative data of the according years in the participating hospitals. The data are owned by the participating hospitals. A list of participating hospitals, and their contact information is included in the supporting information. List of participating hospitals please see S2 Table.

**Funding:** MB (Maria Bogdanovskaya) is a former and LB (Lorna Barclay) a current employee of Dr Foster Telstra Health, London, UK. They performed the analyses of the paper and participated in the design of the study and preparation of the manuscript. Dr Foster Telstra Health was responsible for the collection of data, which were provided by each individual participating hospital. No adjustments to these data were made.

**Competing interests:** MB and LB had an active role in the analysis of the data and participated in the design of the analysis of this manuscript. The company (Dr Foster Telstra Health) collected the data, but did not influence the study design and the data analysis. The company also did not have any influence on the decision to publish or the preparation of the manuscript. The involvement of two employees of the company does not alter our adherence to PLOS ONE policies on sharing data and materials. However, the data cannot be made available publicly as indicated below.

## Conclusion

Readmission in chronic conditions is very common and often caused by diseases other than the index hospitalisation. Interventions to reduce readmissions should therefore focus not only on the primary condition but on a holistic consideration of all the patient's comorbidities.

## Introduction

Chronic diseases are increasingly prevalent in Western countries due to population aging and better treatment of underlying conditions. Once hospitalised, the chance for another hospitalisation increases sharply. The wide variation in readmission rates suggests that a significant proportion is avoidable [1]. The US Hospital Readmissions Reduction Program requires Centers for Medicare and Medicaid Services to reduce payments to hospitals with excess readmissions [2].

Readmissions pose a significant burden to patients and healthcare systems as they create significant mortality and morbidity [3]. The latter is not only important regarding well-being, but also has an important economic impact in form of costly hospitalisations [4]. Therefore, one important aim in treating chronic diseases is to prevent readmissions. The high risk of readmission has been shown for different chronic diseases [5–7] and after diverse interventions such as hip replacement, hip fracture, otolaryngology, bypass or general surgery [8–12]. The reasons for readmission after an intervention are often not primarily related to the intervention itself, but to the underlying comorbid conditions. Thus, chronic diseases may play an important role in readmission risk, independently of the reason for the initial hospitalisation [13].

Chronic diseases usually do not occur in isolation. Most patients with chronic disease have multiple diseases [14]. They may influence each other, and treatment for one disease may adversely impact the other. Hospital quality also impacts readmission rates [15]. Therefore, the pattern of readmissions may be very complex. Because of the interactions between disease and treatment, knowing patterns of readmissions related to different chronic diseases may improve the understanding of this important problem and reveal new treatment modalities for these patients. Unfortunately, most previous studies focused on single disease presentation at index hospitalisation and did not investigate the interplay between different diseases [16].

Therefore, we aimed to investigate reasons for readmission in patients with an index hospitalisation for multiple chronic conditions and three acute diseases usually related to an underlying chronic condition. We used data from an international dataset created from hospitals around the world through the Global Comparators project [17]. Our hypothesis was that reasons for readmission are often different from the index hospitalisation, irrespective of the underlying condition, and that this complexity increases with age and comorbidities. We also wanted to determine if the result changed over time, to what extent the results were influenced by in-hospital mortality, and if there were country specific differences.

## Methods

We collected electronic inpatient records from administrative discharge data provided by each of the 21 participating hospitals (eight from USA, five from UK, four from Australia, one each from Belgium, Denmark, Italy and the Netherlands; the latter four are combined as European centres; S1 Table), integrated into a uniform dataset as described elsewhere [17]. Institutional regulatory boards waived the requirement of Informed consent by patients and data were fully

anonymised for the purpose of this analysis. For the present analysis, we used data from six complete calendar years, 2009–2014. All hospitals were major teaching hospitals. We excluded data before (i.e. 2007–2008) and after (i.e. 2015 and 2016) this period because many centres did not provide data for those periods to avoid selection bias. Between 2011–2013, complete data are available in all centres; in the two years before data from four hospitals (three in USA, one European) and in 2014 data from two UK hospitals were missing. Some hospitals reported in a previous publication [18] were excluded due to data quality issues, some missing codes and linkage issues from one year to the next.

We included records containing information on age, sex, country, principal diagnosis code (International Classification of Diseases, ICD-9 or ICD-10 [ICD-10_CM, ICD-10 AM]), Clinical Classification Software (CCS) group (Agency for Healthcare Quality and Research's CCS groups (AHRQ CCS)) as described previously, [17] admission date, discharge date (date of death if admission ended in death) and in-hospital death. Each record was assigned a comorbidity score according to a modified version of the Elixhauser index [19], which is based on 32 conditions identified by secondary diagnoses codes [20]. Admissions were assigned to one of 259 diagnostic groups based on the primary diagnosis field (AHRQ CCS). Both bespoke chronic diseases (ICD-classification) and CCS codes were used to define the following chronic conditions as index hospitalisations: heart failure (HF); atrial fibrillation (AF); myocardial infarction (AMI); hypertension; stroke; chronic obstructive pulmonary disease (COPD); bacterial pneumonia; diabetes mellitus; chronic renal disease; anaemia; arthritis and other cardiovascular disease (see S2 Table). We included three acute conditions, i.e. stroke, AMI and bacterial pneumonia; AMI because it represents the most important event of an important underlying chronic condition, i.e. coronary artery disease, and pneumonia because it is often related to underlying chronic lung diseases, particularly COPD.

We excluded records corresponding to planned day-cases. Due to the difficulty in some countries in distinguishing patients admitted for observation only from those admitted as inpatients, we also excluded short-term emergency admissions with length of stay (LOS) less than two nights and no surgery.

Rates and reasons for readmissions within 30 days were analysed, focusing on the comparison between similar and different reasons for readmission compared with the index hospitalisation. Descriptive comparisons were made for each of the above-mentioned diseases. For a better illustration, Sankey diagrams are used to show the readmission patterns for each of these conditions [21]. Readmissions could only be identified when readmission was to the same hospital. For the analysis investigating readmission rates, in-hospital deaths were not considered. We allowed for a three-month window at the end of the time-period to ensure capturing the majority of readmissions for each hospital. For consistency across countries, readmissions with discharge after a three-month window were not counted, which accounted for <0.5% of the cases.

We further compared the readmission distributions for each of the 12 diseases, given by the proportion of readmissions to each of the 12 conditions or any CCS group. We calculated the Hellinger distance [22] between each pair, to identify clusters within the readmission patterns of the 12 diseases. By definition, the Hellinger distance is between zero and one, where zero means that patterns are identical, the larger the distance the more dissimilar the patterns.

To investigate the proportion of readmissions with similar versus different diseases we performed logistic regression with the outcome: readmission with the same versus any other condition. The model was adjusted for age, sex, year of discharge, comorbidity score, country, chronic condition and LOS. We applied a square root transformation to LOS as data are highly skewed to the right. Variables were considered significant when p<0.05. All data analyses were done using R, v. 3.4.2 [23].

**Table 1. Patient characteristics.**

|  | Average age | Average number of comorbidities | % male | Average length of stay (LOS) | Median LOS |
|---|---|---|---|---|---|
| **Overall** | 65.4 | 2.29 | 56.6 | 7.5 | 4 |
| **Country** |  |  |  |  |  |
| Australia | 68.4 | 1.81 | 58.1 | 8.5 | 4 |
| Continental Europe | 66.6 | 1.38 | 57.8 | 7.7 | 5 |
| England | 66.3 | 1.98 | 58.6 | 9.3 | 5 |
| USA | 62.5 | 3.30 | 53.6 | 5.8 | 4 |
| **Chronic condition** |  |  |  |  |  |
| AF | 65.6 | 1.80 | 57.0 | 4.5 | 3 |
| AMI | 66.9 | 2.26 | 69.9 | 5.7 | 3 |
| Anemia | 52.4 | 1.68 | 45.3 | 6.2 | 4 |
| Arthritis | 63.9 | 1.46 | 42.1 | 4.3 | 3 |
| COPD | 68.6 | 1.77 | 51.0 | 7.6 | 5 |
| Diabetes | 52.4 | 2.20 | 56.5 | 7.8 | 4 |
| Heart failure | 70.6 | 3.57 | 55.5 | 8.9 | 6 |
| Hypertension | 62.4 | 1.49 | 43.3 | 4.9 | 3 |
| Other cardiovascular | 67.7 | 2.10 | 60.9 | 8.5 | 4 |
| Pneumonia | 64.2 | 2.31 | 53.3 | 8.9 | 6 |
| Renal failure | 63.1 | 3.06 | 56.7 | 8.5 | 5 |
| Stroke | 68.5 | 2.53 | 50.3 | 12.7 | 6 |

Abbreviations: AF atrial fibrillation; AMI acute myocardial infarction; COPD chronic obstructive pulmonary disease

## Results

Out of 4,901,586 admissions, 866,502 (17.7%) were due to the 12 bespoke conditions as primary diagnosis (UK n = 207,748, 15.4%; continental Europe n = 187,182, 17.7%; Australia n = 161,171, 18.0%; USA n = 310,401, 19.4% USA). The patient characteristics are shown in Table 1. Patients in the USA were younger compared with the other countries, had a shorter LOS and more co-morbidities recorded. There were significant differences in baseline characteristics between the different conditions. Average LOS varied by a factor of almost three between the different conditions and differed between countries (Table 1). Table 2 shows the most common CCS groups / chronic conditions by number of admissions, indicating that the selected conditions are among the most important reasons for hospitalisation. Fig 1 provides an overview of the reasons of admission in the different countries over time.

Overall, 43,573 (5.0%) patients died in-hospital with significant differences between the conditions (Table 3), leaving 822,929 admissions for readmission analysis. Of those, 87,452 (10.6%) had an emergency readmission within 30 days (Table 3). Readmission rate was highest in the USA (19.0%), followed by Australia (17.6%), England (16.5%) and continental Europe (14.6%). No major changes were seen over time, but there were some differences between the countries (Fig 2). Apart from anaemia and COPD, more than half of the readmissions were due to diseases other than the initial hospitalisation (Table 3). Interestingly, average LOS for the initial admission was comparable between patients that were not readmitted and were readmitted with the same condition in all 12 conditions but significantly longer in those readmitted due to another condition (Fig 3). Though the extent of this increase in LOS varied, it was seen in basically all conditions.

Overall, the average and median time to readmission did not differ between readmission due to the same or another condition (Fig 4). The median time until readmission was 11 days

**Table 2. Most common CCS groups and chronic conditions as primary cause of initial hospitalization.**

| Condition group | Volume | Percentage |
|---|---|---|
| CCS group Liveborn | 231377 | 4.72% |
| Chronic condition AMI | 175437 | 3.58% |
| CCS group Complication of device, implant or graft | 115533 | 2.36% |
| Chronic condition Pneumonia | 97496 | 1.99% |
| CCS group Other complications of birth, puerperium affecting management of mother | 90551 | 1.85% |
| Chronic condition Stroke | 86189 | 1.76% |
| Chronic condition Heart failure | 84963 | 1.73% |
| Chronic condition AF | 82723 | 1.69% |
| CCS group Complications of surgical procedures or medical care | 80353 | 1.64% |
| CCS group Spondylosis, intervertebral disc disorders, other back problems | 75086 | 1.53% |
| Chronic condition Other cardiovascular disease | 72414 | 1.48% |
| CCS group Other nervous system disorders | 66199 | 1.35% |
| CCS group Biliary tract disease | 65289 | 1.33% |
| Chronic condition COPD | 63450 | 1.29% |
| CCS group Secondary malignancies | 61933 | 1.26% |
| CCS group Urinary tract infections | 61488 | 1.25% |
| CCS group Residual codes, unclassified | 60878 | 1.24% |
| Chronic condition Arthritis | 59969 | 1.22% |
| CCS group Septicemia (except in labour) | 59506 | 1.21% |
| Chronic condition Chronic renal failure | 57209 | 1.17% |
| CCS group Skin and subcutaneous tissue infections | 56959 | 1.16% |
| CCS group Trauma to perineum and vulva | 52616 | 1.07% |
| CCS group Normal pregnancy and/or delivery | 51040 | 1.04% |
| CCS group Maintenance chemotherapy, radiotherapy | 50700 | 1.03% |
| CCS group Fracture of upper limb | 49437 | 1.01% |
| CCS group Foetal distress and abnormal forces of labour | 48947 | 1.00% |
| CCS group Rehabilitation care, fitting of prostheses, and adjustment of devices | 46808 | .95% |
| CCS group Fracture of lower limb | 46628 | .95% |
| CCS group Other and unspecified benign neoplasm | 45375 | .93% |
| CCS group Epilepsy, convulsions | 44791 | .91% |
| CCS group Osteoarthritis | 44689 | .91% |
| CCS group Abdominal hernia | 44214 | .90% |
| CCS group Other connective tissue disease | 44197 | .90% |
| CCS group Other complications of pregnancy | 44190 | .90% |
| Chronic condition Diabetes | 41136 | .84% |
| Chronic condition Anemia | 38163 | .78% |
| CCS group Heart valve disorders | 38089 | .78% |
| CCS group Affective disorders | 37311 | .76% |
| CCS group Fracture of neck of femur (hip) | 37177 | .76% |
| CCS group Other fractures | 36657 | .75% |
| CCS group Appendicitis and other appendiceal conditions | 36167 | .74% |
| CCS group Other gastrointestinal disorders | 34775 | .71% |
| CCS group Other perinatal conditions | 34214 | .70% |
| CCS group Crushing injury or internal injury | 33283 | .68% |
| CCS group Other upper respiratory disease | 32306 | .66% |
| CCS group Cancer of bronchus, lung | 31956 | .65% |
| CCS group Gastrointestinal haemorrhage | 31834 | .65% |

(*Continued*)

**Table 2.** (Continued)

| Condition group | Volume | Percentage |
|---|---|---|
| CCS group Abdominal pain | 31453 | .64% |
| CCS group Cancer of breast | 31219 | .64% |
| CCS group Intracranial injury | 29403 | .60% |
| CCS group Intestinal obstruction without hernia | 29162 | .59% |
| CCS group Fluid and electrolyte disorders | 28882 | .59% |
| CCS group Other nutritional, endocrine, and metabolic disorders | 28152 | .57% |
| CCS group Intestinal infection | 27751 | .57% |
| CCS group Pancreatic disorders (not diabetes) | 27448 | .56% |
| CCS group Calculus of urinary tract | 27337 | .56% |
| CCS group Acute bronchitis | 27261 | .56% |
| CCS group Nonspecific chest pain | 26377 | .54% |
| CCS group Other screening for suspected conditions | 24198 | .49% |
| CCS group Asthma | 24019 | .49% |

Abbreviations: AF atrial fibrillation; AMI acute myocardial infarction; CCS Clinical Classification Software (definition see Methods); COPD chronic obstructive pulmonary disease

for both the same and another condition. Average time until readmission did not vary much between the 12 conditions with only hypertension and stroke having considerably shorter time to readmission for the same condition (Fig 4).

The specific reasons for readmission of the 12 conditions are depicted in Fig 5A and 5B. These figures show that the causes of readmissions for a reason other than the same condition are diverse for all conditions. Regarding other causes than the 12 conditions, complications related to procedures / care during the initial hospitalisation or related to devices / implants are the most common ones, followed by infections. However, there was a large diversity of other causes (Fig 5B).

In addition to readmission for the same reason (Hellinger distance = 0), several weak clusters (light orange) were identified, such as pneumonia and COPD; HF, MI and AF; other cardiovascular, chronic renal failure and arthritis (Fig 6). COPD and anaemia had the greatest distance from each other, and anaemia was far away from all other conditions. Anaemia was also the condition with the highest proportion of readmissions with the same condition (Table 3).

Various factors made readmission with the same condition more or less likely (Table 4). Patients with the initial hospitalisation due to COPD, HF, anaemia, and diabetes were most likely admitted with the same condition. In the USA, readmission due to the same cause was less likely. The chance of being readmitted due to the same cause was increased in male patients, with lower age, with less co-morbidities, at the beginning of the study period and with shorter LOS during initial admission. The effect of age was particularly evident in patients aged <60 years, and less evident at older ages, but the effect of age was not the same for all chronic conditions (Fig 7).

## Discussion

In this large multinational cohort of hospital admissions with various chronic conditions, readmissions within 30 days were common and the reason for readmission differed from the index admission in more than 50% of the cases. This is true among all participating countries and all conditions investigated, despite significant differences between countries and

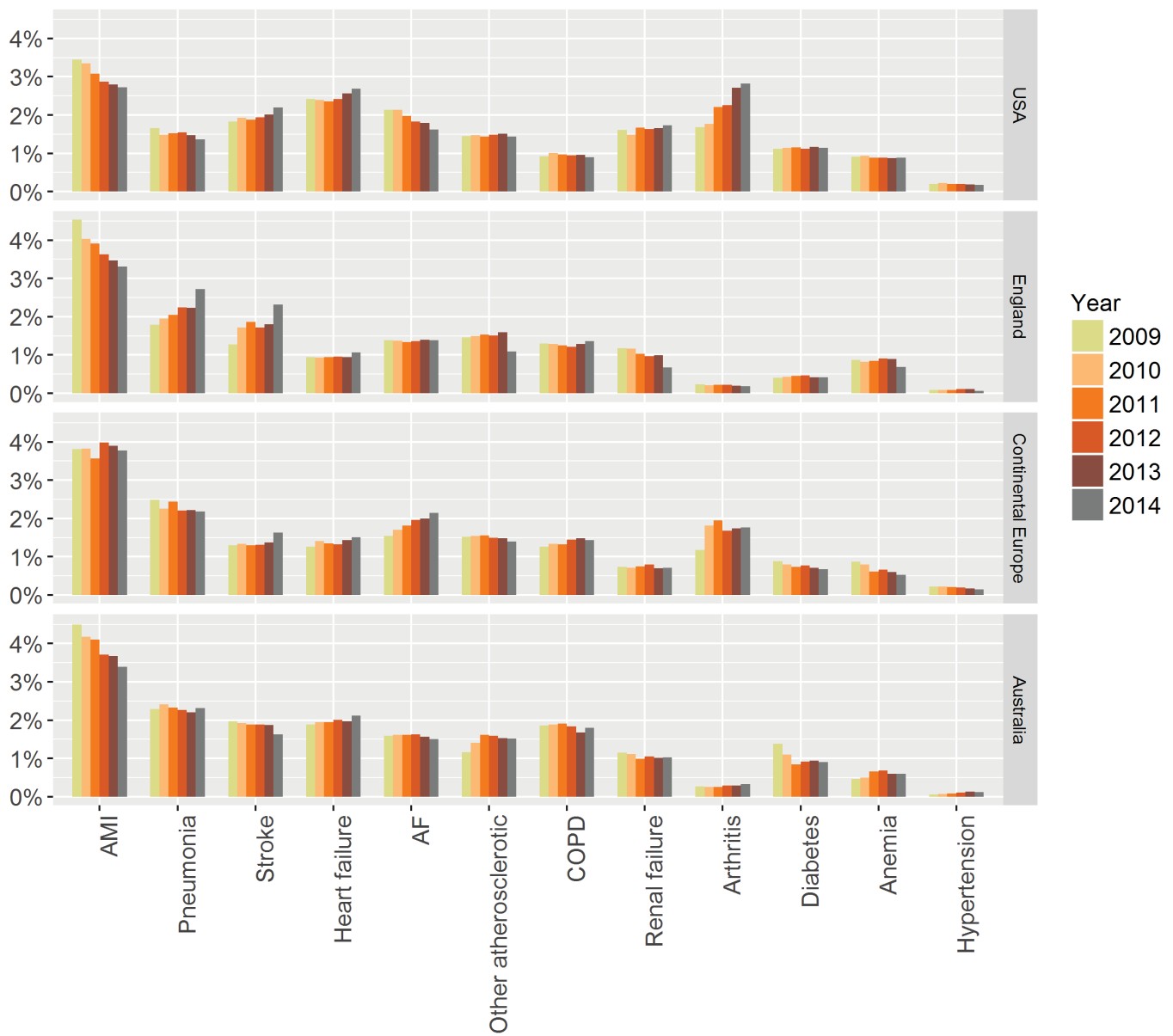

**Fig 1. Proportion of admissions over time for the 12 chronic conditions in different countries.** Abbreviations: AF atrial fibrillation; AMI acute myocardial infarction; COPD chronic obstructive pulmonary disease.

conditions. This is the most comprehensive study reporting readmission rates for a broad spectrum of chronic conditions in different countries. It highlights the extent and the complexity of the problem as well as the need for focus not only on the disease that caused the initial hospitalisation, but also on other frequently related conditions. This stresses the need to approach readmission prevention from a holistic consideration of all the patient's comorbidities.

When a person-centred approach is to the whole patient, readmission rates may be reduced [24]. It is surprising that there is not more focus on this, given that the number of comorbidities is associated with increased mortality and an increased readmission rate, both early and

**Table 3. Rate of in-hospital mortality, 30-day readmissions, percentage of conditions of readmissions and total events (combination of readmissions and deaths) in relation to the reason for index hospitalisation.**

| Condition | Total admissions | # of deaths | In-hospital mortality | Total discharges | 30-day readmissions | % of all readmissions | | | Total events | Percentage events |
|---|---|---|---|---|---|---|---|---|---|---|
| | | | | | | Same condition | Other chronic condition | Other condition | | |
| Pneumonia | 97,496 | 9,959 | 10.2% | 87,537 | 11,507 (13.1%) | 24.1% | 20.5% | 55.4% | 21,466 | 22.0% |
| Heart failure | 84,963 | 5,078 | 6.0% | 79,885 | 13,333 (16.7%) | 42.1% | 22.7% | 35.2% | 18,411 | 21.7% |
| COPD | 63,450 | 2,456 | 3.9% | 60,994 | 11,231 (18.4%) | 51.9% | 17.9% | 30.2% | 13,687 | 21.8% |
| Renal failure | 57,209 | 3,101 | 5.4% | 54,108 | 8,579 (15.9%) | 18.4% | 18.8% | 62.8% | 11,680 | 20.4% |
| Anemia | 38,163 | 568 | 1.5% | 37,595 | 6,818 (18.1%) | 52.9% | 10.2% | 36.9% | 7,386 | 19.4% |
| Stroke | 86,189 | 11,105 | 12.9% | 75,084 | 5,134 (6.8%) | 26.8% | 14.8% | 58.4% | 16,239 | 18.8% |
| Diabetes | 41,136 | 580 | 1.4% | 40,556 | 5,310 (13.1%) | 42.6% | 13.8% | 43.6% | 5,890 | 14.3% |
| Other cardiovascular | 72,414 | 3,894 | 5.4% | 68,520 | 6,433 (9.4%) | 21.2% | 17.6% | 61.2% | 10,327 | 14.3% |
| Hypertension | 7,353 | 63 | 0.9% | 7,290 | 620 (8.5%) | 22.9% | 35.7% | 41.4% | 683 | 9.3% |
| AMI | 175,437 | 5,374 | 3.1% | 170,063 | 10,659 (6.3%) | 27.4% | 27.4% | 45.2% | 16,033 | 9.1% |
| AF | 82,723 | 1,326 | 1.6% | 81,397 | 6,169 (7.6%) | 32.2% | 26.3% | 41.5% | 7,495 | 9.1% |
| Arthritis | 59,969 | 69 | 0.1% | 59,900 | 1,659 (2.8%) | 10.9% | 17.4% | 71.7% | 1,728 | 2.9% |
| Total | 866,502 | 43,573 | 5.0% | 822,929 | 87,452 (10.6%) | 33.9% | 19.9% | 46.2% | 131,025 | 15.1% |

Abbreviations: AF atrial fibrillation; AMI acute myocardial infarction; COPD chronic obstructive pulmonary disease

later after discharge [25, 26]. If comorbidities primarily increase the susceptibility of deteriorating with the same condition or simply increase the risk of manifestation of other diseases has not yet been properly investigated, but comorbidities clearly increase the complexity of the condition [27]. In patients with an index hospitalisation of COPD, readmission due to other causes was accompanied with significantly higher mortality than readmission due to COPD [28]. Therefore, the patterns accompanied with the highest risks of readmission should be more adequately investigated and addressed to prevent early events. Readmission is a multifactorial phenomenon constructed as a quality measure in many countries, with contributions from underlying disease and comorbidities (as found in our data), patient factors (e.g. psychosocial resilience), healthcare worker effects, environmental and social determinants of health, organisational and healthcare system factors [15, 16, 29, 30]. In addition, the organisation of care post-discharge, such as timeliness of follow-up, coordination with primary care, and quality of medication management may significantly influence readmission rate [31, 32]. Preventable readmission is what we should be focusing on, but consensus on how to achieve this is poor [33]. Our data suggest that the focus on and treatment of the index condition only, as currently done in many institutions is not sufficient. Thus, our results explain, at least in part, why measures to prevent rehospitalisation have had only a limited impact on the readmission rate, and that successful programs must address multiple aspects of patient care [26, 30].

Disease management programs are advocated in chronic conditions such as HF or COPD after discharge to reduce the readmission rate [34–36]. Although there is no uniform way to provide managed care, the majority of such programs improve outcome as shown in several meta-analyses [37–39]. The rather broad approach to care of these programs might result in reduction of readmission not only due to the index disease. The clustering of diseases with some clinical validity found in our analysis might allow such assumption. This notion remains speculative, however, until properly tested.

Although the spectrum of causes for readmissions was broad, independently of the cause of the initial admission, there are some conditions that were more common. Thus, myocardial

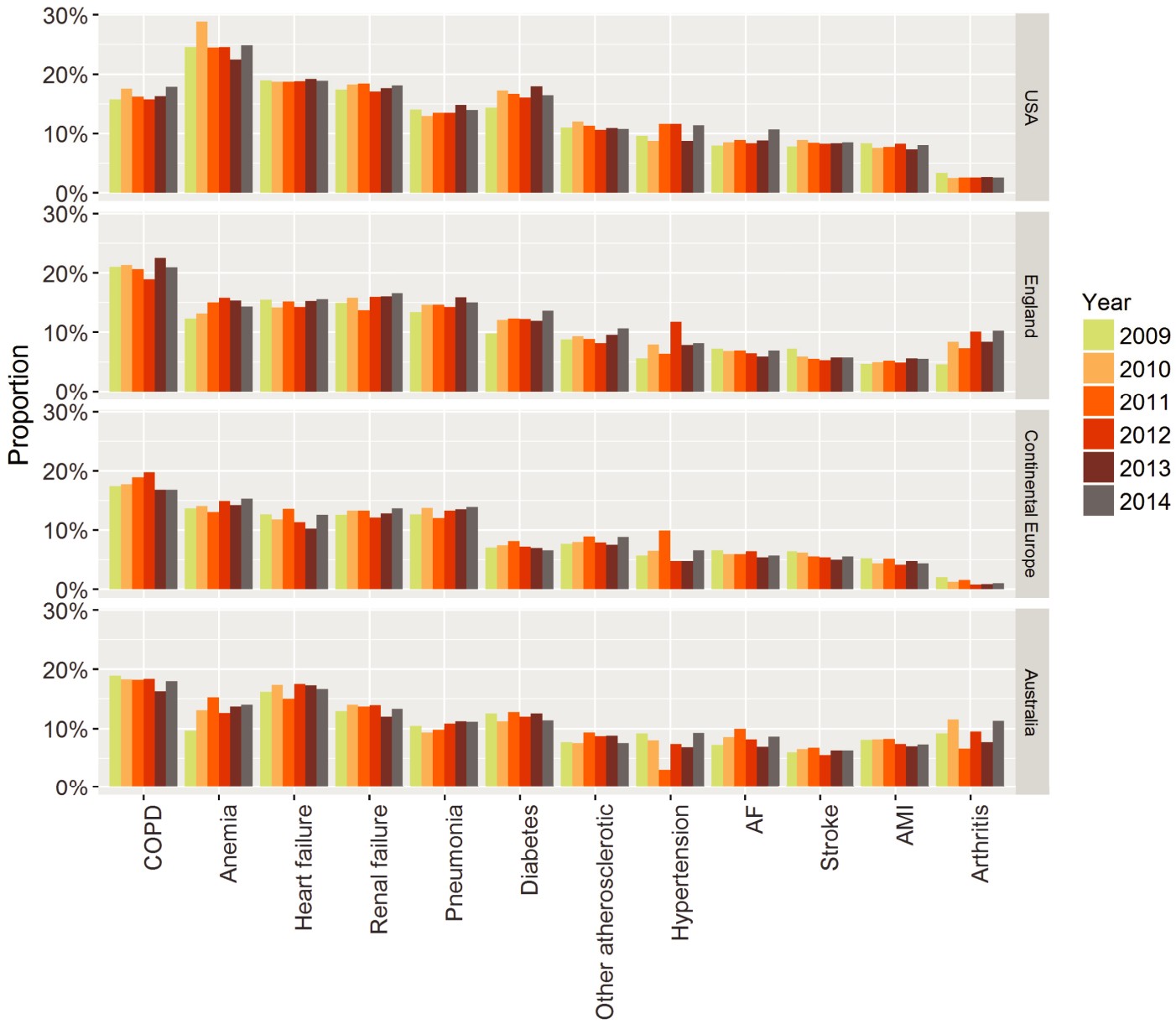

**Fig 2. Proportion of readmissions over time in different countries for each of the 12 chronic conditions.** Abbreviations: AF atrial fibrillation; AMI acute myocardial infarction; COPD chronic obstructive pulmonary disease.

infarction or atrial fibrillation may lead to HF, (treatment of) HF may cause renal failure, and patients with COPD are more susceptible to acquire pneumonia. There were, however, causes that were seen in basically all conditions. Many patients had infections as cause of readmission, in line with previous reports [28]. It suggests that such patients are vulnerable early after discharge to acquire infectious diseases or infections may be acquired during hospitalisation (e.g. urinary tract infections due to urine catheters; hospital acquired pneumonia). Additionally, the susceptibility of these patients may be higher than during more stable conditions. Using

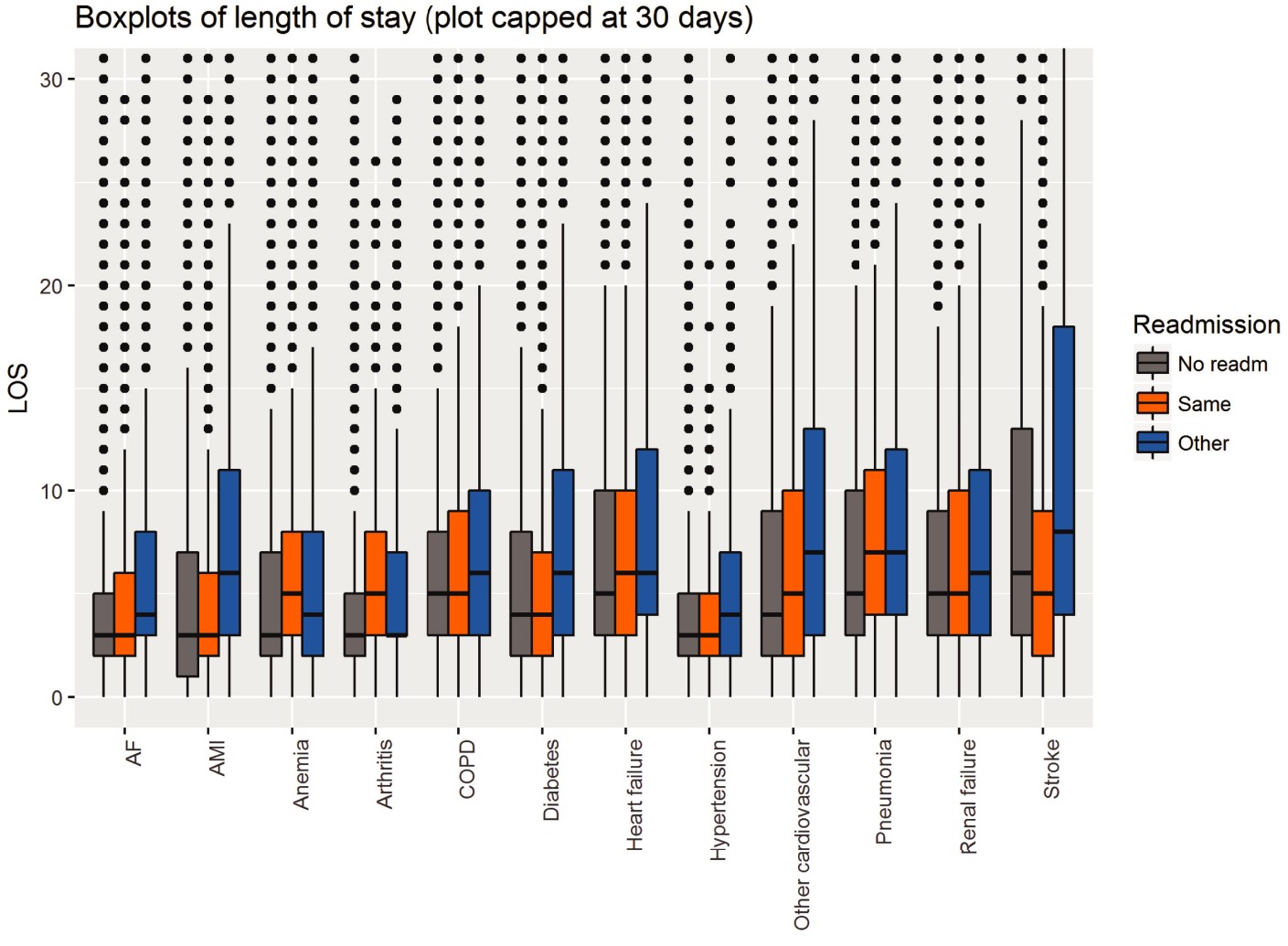

**Fig 3. Length of stay (LOS) for the index hospitalisation of the chronic conditions in those that were not readmitted (grey), readmitted with the same condition (orange) and with another condition (blue).** Abbreviations: AF atrial fibrillation; AMI acute myocardial infarction; COPD chronic obstructive pulmonary disease.

sufficient hygiene measures such infections might in part be preventable if the general awareness is improved. Education of patients and their families about the increased risk of infection may also be important to aid rapid diagnosis and treatment.

Another important readmission group is related to complications of medical procedures or devices. Although it is not well investigated to what extent such complications are preventable in the setting of chronic conditions, it may be worthwhile to specifically investigate such complications and to test preventive measures to reduce the number. Often, such measures are only taken in the context of surgery and/or invasive procedures, but not as a general routine in patients with chronic conditions.

There was a small, but significant trend for less readmissions with the same condition as the index hospitalisation over time. The aging population and the fact that comorbidities are increasingly frequent at higher age may be accountable for this finding. Age and comorbidities are included in the regression equation as well, which may highlight this, but may have

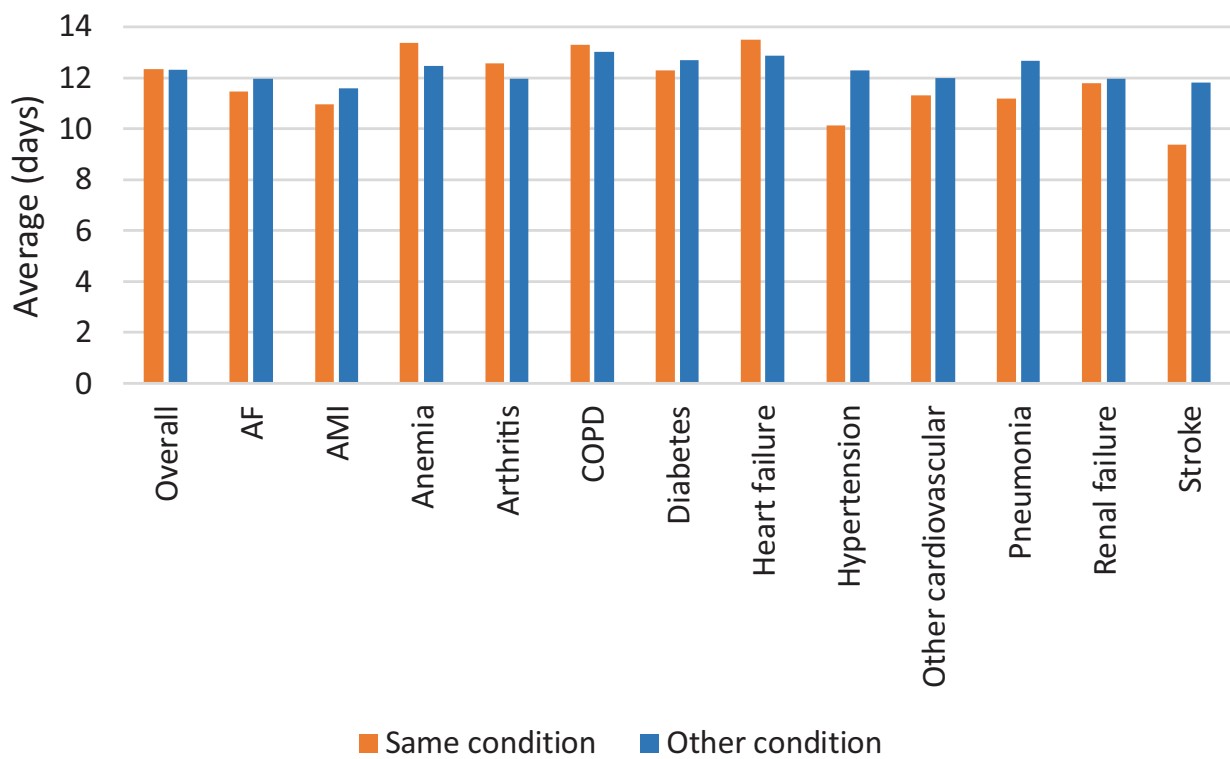

**Fig 4. Average time to readmission overall and per original condition, depending on if readmission was due to the same or another condition.**
Abbreviations: AF atrial fibrillation; AMI acute myocardial infarction; COPD chronic obstructive pulmonary disease.

diminished the overtime effect. Therefore, we may expect that the complexity of patients admitted with chronic conditions is increasing in the future as the population ages further. It may also be explained as less complex procedures, and less severe conditions, are treated more in the outpatient setting. Therefore, the increasing complexity of hospitalised patients needs to be considered for future planning of hospital care of chronic diseases.

## Limitations

There are several limitations to our study. Importantly, we used administrative data, capturing only what was coded. Coding is not uniform and may vary depending on diagnosis related group, degree of in-hospital testing, national reimbursement incentives, physician and coder training and institutional variation [40, 41]. Some of these factors may explain why patients in the USA had a higher comorbidity score although they were younger and had a shorter LOS. The higher comorbidities in US patients could also reflect issues of access and late presentation [42]. Direct comparison between countries, and also between institutions, regarding the absolute burden of disease is not possible. Also, risk adjustment considering all factors influencing readmissions is limited. However, the purpose of this study was to investigate the associations between index hospitalisation and readmission. As this only relates to the primary diagnosis, coding has a much lower impact. Moreover, the aim was to investigate the global picture of the problem, which is not significantly influenced by the limitation of coding. We did not compare single institutions with each other due to reasons of confidentiality given the limited number of participating hospitals, particularly in continental Europe. Administrative models may have

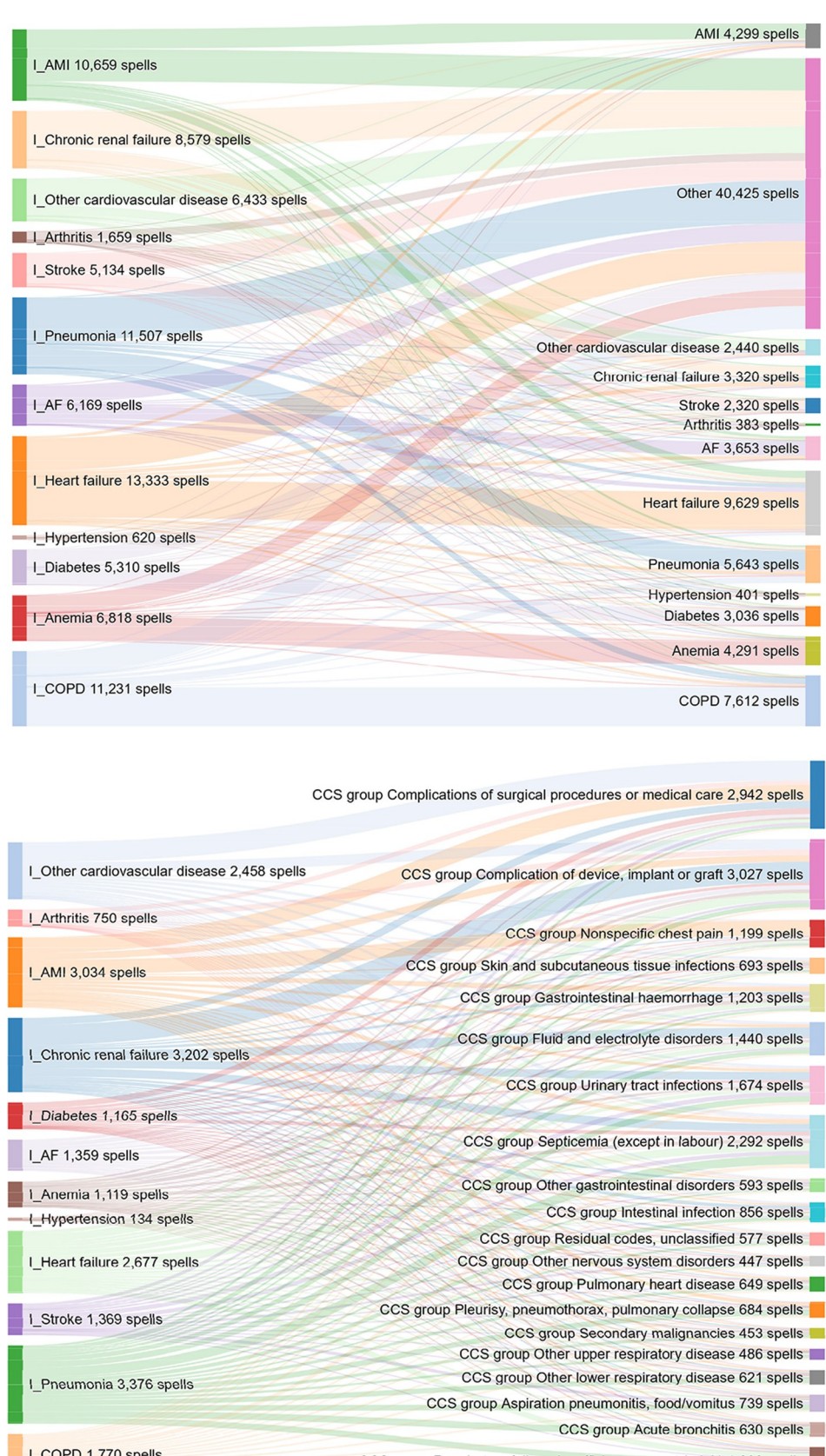

**Fig 5. Link between original condition of admission (left side) and cause of readmission (right side) for each chronic condition.** B provides information about all other causes in A. These figures (Supporting information: Interacting Fig 1A and 1B) can be found online by clicking on the figures to provide a dynamic depiction of the different causes of readmission for each chronic condition. Abbreviations: AF atrial fibrillation; AMI acute myocardial infarction; COPD chronic obstructive pulmonary disease.

limited discriminatory abilities, which raises concerns about the ability to standardise risk across hospitals to fairly compare hospital performance [16]. Our data suggest that previously insufficiently recognised factors related to the complexity of multimorbid patients need to be included to improve risk prediction and adjustment.

**Fig 6. Readmission pattern for each of the chronic conditions (Hellinger distance matrix).** Abbreviations: AF atrial fibrillation; AMI acute myocardial infarction; COPD chronic obstructive pulmonary disease.

**Table 4. Odds ratios (OR) of multivariable regression for probability of readmission with the same condition as initial hospitalisation.**

|  | OR (95%-confidence interval) |
|---|---|
| Chronic Condition: Arthritis | 1.000 (Reference) |
| Chronic Condition: AF | 4.257*** (3.610, 5.021) |
| Chronic Condition: AMI | 3.366*** (2.863, 3.958) |
| Chronic Condition: Anaemia | 7.328*** (6.219, 8.634) |
| Chronic Condition: Renal failure | 2.021*** (1.713, 2.386) |
| Chronic Condition: COPD | 9.460*** (8.057, 11.109) |
| Chronic Condition: Diabetes | 5.468*** (4.632, 6.454) |
| Chronic Condition: Heart failure | 7.774*** (6.624, 9.122) |
| Chronic Condition: Hypertension | 2.485*** (1.943, 3.179) |
| Chronic Condition: Other cardiovascular | 2.397*** (2.027, 2.834) |
| Chronic Condition: Pneumonia | 2.934*** (2.494, 3.451) |
| Chronic Condition: Stroke | 3.852*** (3.255, 4.559) |
| Country: Australia | 1.000 (Reference) |
| Country: England | .968 (.922, 1.017) |
| Country: Continental Europe | .979 (.935, 1.026) |
| Country: USA | .728*** (.698, .760) |
| Female | .954*** (.926, .983) |
| Age | .984*** (.983, .984) |
| Comorbidity score | .986*** (.985, .987) |
| Year | .987*** (.978, .996) |
| LOS (sqrt days) | .870*** (.859, .881) |
| C-statistic | .695 |
| Observations | 87,452 |

*p < .1
**p < .05
***p < .01
Abbreviations: AF atrial fibrillation; AMI acute myocardial infarction; COPD chronic obstructive pulmonary disease

We included conditions that present with acute events (i.e. AMI, pneumonia, stroke) but are usually caused by a chronic underlying disease. Including these conditions might have influenced our results. However, readmission pattern of them were in line with the other more chronic conditions, supporting the assumption that the readmissions were mainly influenced by the underlying chronic condition rather than the acute event.

In addition, participating hospitals are academic centres and focus may differ between centres. Therefore, there may be an imbalance in the patient population and differences in the spectrum of patients in centres that did not participate. Also, readmission to a different hospital could not be recorded. Therefore, the true readmission rate is likely even higher than reported here although the disease specific readmission rates may be similar to those reported in the past (for example 7 vs 5% for heart failure) [43]. It might be speculated that readmission due to other causes than the initial admission is more likely to be treated in another hospital than the index admission and the absolute values must be interpreted with caution. Nevertheless, the consequence of these two factors would be an even higher readmission rate and a higher proportion of causes different than the initial admission, even more strongly supporting the conclusion of our study.

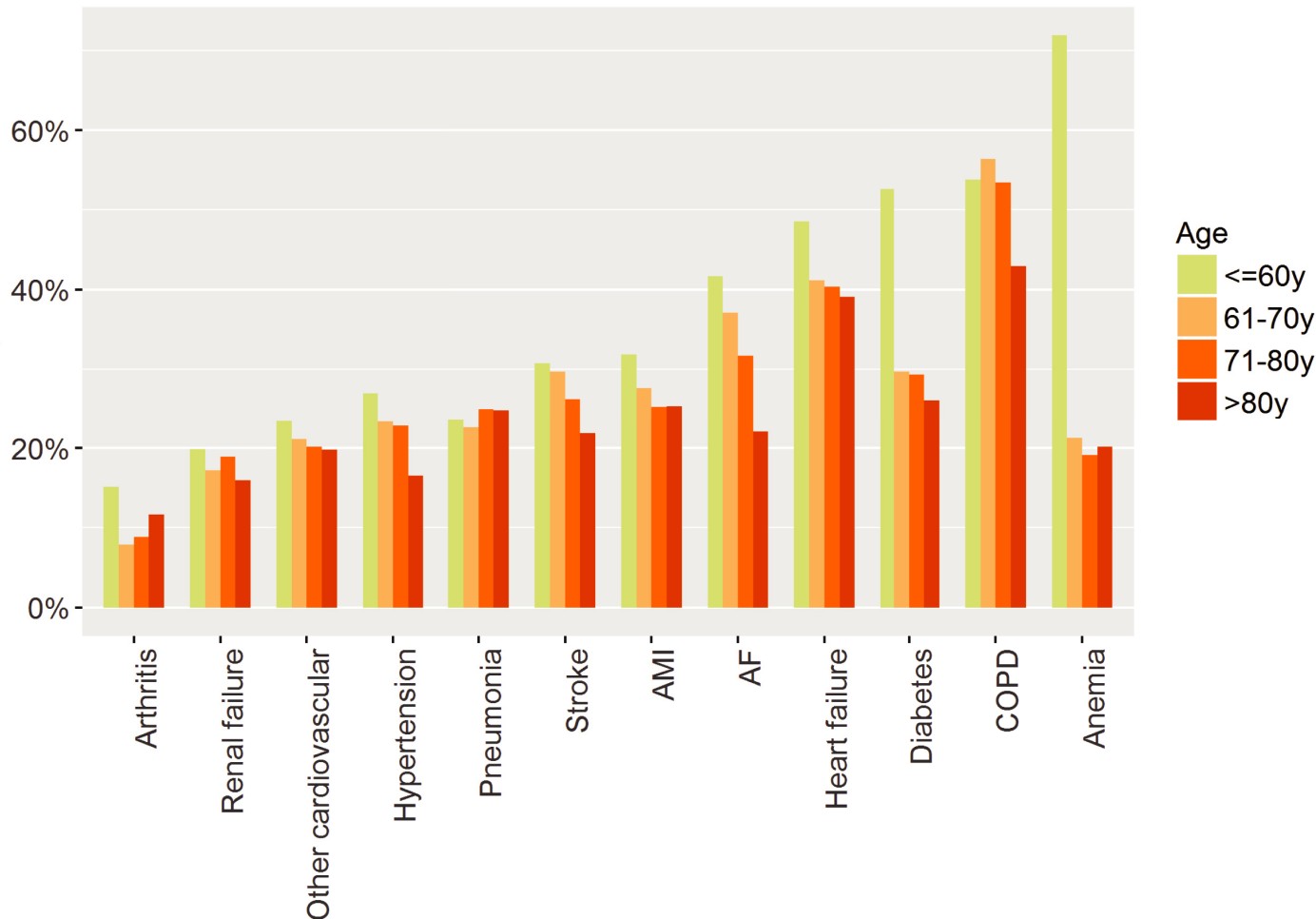

**Fig 7. Readmission rates with the same condition by age for each of the chronic conditions.** Abbreviations: AF atrial fibrillation; AMI acute myocardial infarction; COPD chronic obstructive pulmonary disease.

## Conclusions

Readmissions within 30 days after discharge are very common in patients with chronic conditions. These readmissions are, in more than 50% of patients, related to a different cause than the initial hospitalisation. Therefore, this study has significant implications, for clinicians, the hospital administrators and for policy makers. It highlights the importance of a global approach to the treatment and management of patients with chronic conditions. Focus on the chronic condition and the circumstances that led to the hospitalisation is not sufficient, and additional measures based on a more holistic approach to the individual patient should be taken to significantly reduce readmissions.

## Supporting information

**S1 Table. List of participating centres.**
(DOCX)

**S2 Table. ICD 9 and ICD 10 codes for chronic conditions.**
(DOCX)

**S1 Fig.** A and B: Link between original condition of admission (left side) and cause of read-mission (right side) for each chronic condition. B provides information about all other causes in A. Interactive online figures to provide a dynamic depiction of the different causes of read-mission for each chronic condition.
(HTML)

## Acknowledgments

Mrs Maria Bogdanovskaya is former and Dr Lorna Barclay current employee of Dr Foster Telstra Health, London, UK.

## Author Contributions

**Conceptualization:** Hans-Peter Brunner-La Rocca, Carol J. Peden, John Soong, Per Arne Holman, Lorna Barclay.

**Data curation:** Maria Bogdanovskaya.

**Formal analysis:** Maria Bogdanovskaya, Lorna Barclay.

**Methodology:** Hans-Peter Brunner-La Rocca, Carol J. Peden, John Soong, Per Arne Holman, Maria Bogdanovskaya, Lorna Barclay.

**Supervision:** Hans-Peter Brunner-La Rocca.

**Validation:** Hans-Peter Brunner-La Rocca, Lorna Barclay.

**Visualization:** Hans-Peter Brunner-La Rocca.

**Writing – original draft:** Hans-Peter Brunner-La Rocca.

**Writing – review & editing:** Carol J. Peden, John Soong, Per Arne Holman, Maria Bogdanovskaya, Lorna Barclay.

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
