## [Decision Letter · Decision Letter 0]

24 Feb 2020

PONE-D-19-35626

Reasons for readmission after hospital discharge in patients with chronic diseases  - information from an international dataset

PLOS ONE

Dear Dr. Brunner-La Rocca,

Thank you for submitting your manuscript to PLOS ONE. After careful consideration, we feel that it has merit but does not fully meet PLOS ONE’s publication criteria as it currently stands. Therefore, we invite you to submit a revised version of the manuscript that addresses the points raised during the review process.

We would appreciate receiving your revised manuscript by Apr 09 2020 11:59PM. To enhance the reproducibility of your results, we recommend that if applicable you deposit your laboratory protocols in protocols.io, where a protocol can be assigned its own identifier (DOI) such that it can be cited independently in the future. For instructions see: http://journals.plos.org/plosone/s/submission-guidelines#loc-laboratory-protocols

We look forward to receiving your revised manuscript.

Kind regards,

Gianluigi Savarese

Academic Editor

PLOS ONE

Journal Requirements:

2. In ethics statement in the manuscript and in the online submission form, please provide additional information about the database used in your retrospective study. Specifically, please ensure that you have discussed whether all data were fully anonymized before you accessed them and/or whether the IRB or ethics committee waived the requirement for informed consent. If patients provided informed written consent to have their data used in research, please include this information.

"Maria Bogdanovskaya: I have read the journal's policy and the authors of this manuscript have the following competing interests: former employee of Dr Foster Telstra Health.

Lorna Barclay: I have read the journal's policy and the authors of this manuscript have the following competing interests: employee of Dr Foster Telstra Health."

We note that one or more of the authors are employed by a commercial company: Dr Foster Telstra Health.

Reviewers' comments:

Reviewer's Responses to Questions

**Comments to the Author**

1. Is the manuscript technically sound, and do the data support the conclusions?

Reviewer #1: Yes

Reviewer #2: Partly

Reviewer #3: Yes

2. Has the statistical analysis been performed appropriately and rigorously? 

Reviewer #1: Yes

Reviewer #2: Yes

Reviewer #3: I Don't Know

3. Have the authors made all data underlying the findings in their manuscript fully available?

Reviewer #1: Yes

Reviewer #2: Yes

Reviewer #3: No

4. Is the manuscript presented in an intelligible fashion and written in standard English?

Reviewer #1: Yes

Reviewer #2: Yes

Reviewer #3: Yes

5. Review Comments to the Author

Reviewer #1: The authors study the reasons for readmission after hospital discharge in patients with chronic diseases in an international dataset.

In a very large sample size of 4,901,584 admissions, the authors investigated reasons for readmission in patients with an index hospitalization for multiple chronic conditions and two acute diseases usually related to an underlying chronic condition.

The hypothesis of the authors was that reasons for readmission are often different from the index hospitalisation, irrespective of the underlying condition, and that this complexity increases with age and comorbidities. In addition, the authors determined if the results changed over time.

The study is relevant, interesting and well written. I think this paper has the ability for a better understanding of the treatment and management of patients with chronic conditions and the described high rate of about 17% of readmissions. The conclusions could be strengthened. However, the present study is significantly limited by the administrative nature of the data, the lack of robust risk adjustment and the imbalances in the patient population.

There are some major concerns and limitations which have to be to addressed:

- The authors should clarify whether the identified ICD codes were main or secondary diagnoses for readmission of the patients. This fact could potentially bias the observed findings.

- Why did the authors choose the 30 days as time point of emergency readmission?

- The authors should stratify their findings concerning AMI as reason of readmission in the subtypes of AMI and by OPS codes for coronary interventions. These findings should also be incorporated in the rates of in-hospital mortality and might show differences over the time and by area (country).

- Table 4 shows the calculated odds-ratios of regression for probability of readmission with the same condition as initial hospitalisation. The authors should adjust the findings i.e. by area (country), length of stay and year of admission.

Reviewer #2: In the present paper, Brunner-La Rocca and Colleagues aimed to "investigate reasons for readmission in patients with an index hospitalisation for multiple chronic conditions and two acute diseases usually related to an underlying chronic condition”. The Authors collected electronic inpatient records from administrative discharge data provided by each of the 21 participating hospitals in the years 2009-2014. The final analysis was performed on around 866k admissions due to 12 different chronic conditions. 30-day readmission rates ranged from 2.8% for arthritis to 18.4% for COPD; the Authors also report a high incidence of re-admission due to other (non-chronic) conditions, most commonly, treatment related complications and infections.

The issue of readmission rate in chronic disease is of outmost social and economic importance, but it has been often regarded from a disease-specific point of view. The Authors should be commended for their attempt to provide a picture of the possible paths patients with major chronic conditions often follow, using a huge amount of administrative data from all over the world.

While of interest, the value of the findings of the paper is limited from some major methodological issues, as reported below in details.

Major points

- The Authors state that they “included two acute conditions, i.e. AMI and bacterial pneumonia”. Indeed, stroke (acute?!) was also considered in the analysis. Although they were apparently included as they underly other chronic conditions, the Reviewer is rather concerned about this choice.

- Only readmissions at the same hospital could be recorded. Although the disease specific readmission rates are similar to those reported in the past (for example 7 vs 5% for heart failure, Fudim M et al. Aetiology, timing and clinical predictors of early vs. late readmission following

index hospitalization for acute heart failure: insights from ASCEND-HF. Eur J Heart Fail. 2018;20:304-314), the whole number of readmission may have been significantly underestimated, especially as concerns those caused by different diseases (as patients may have been referred to different specialized hospitals).

- Administrative and reimbursement issues may have significantly influenced the attribution of the readmission cause, thus possibly biasing the Authors’ findings.

- How was time to first readmission related to the readmission cause?

Minor points

- How do the Authors explain their observation that readmission rated have not changed significantly along the whole study period for most of the chronic conditions they have considered?

- Discussion should be shortened and focused on the main findings of the study.

Reviewer #3: Thank you for the opportunity to review this manuscript. It explored causes of admission and readmission in a unique population of multiple academic centers across the world. In a type of consortium 20+ centers have pooled administrative data for purposes of quality improvement research. IN this analysis the researchers found that in most cases the cause of readmission is different from the cause of admission the first time around. These findings have been previously described in a variable type of cohorts including administrative databases and clinical trial cohorts (see my last comment). Nevertheless this data remains very important and only underlines the need for a more “holistic” patient care (as the authors called it).

Comments:

- This study encompasses multiple countries. US policy on readmission reduction was cited but what about other countries? Do they have comparable initiatives?

- I am not sure I follow the difference between ICD and CCS. Aka Table 1 and 2.

- It is important to understand where the ICD code for the primary diagnosis was derived from. Did the ICD code get obtained from the billing information, discharge notes etc? Only the code I the first position was used?

- Found the amount of information provide in the figures to be somewhat overwhelming.

- Abbreviations in all tables and figures need to be presented in legends.

- Information in the discussion section is repetitive. Reduce repetition and reduce in length.

- Given that HF is a major contributor to the admission and readmission consider to discuss some of the already published literature on this topic. PMID: 27133201 and PMID: 29082629.

6. PLOS authors have the option to publish the peer review history of their article (what does this mean?). If published, this will include your full peer review and any attached files.

Reviewer #1: No

Reviewer #2: No

Reviewer #3: No

---

## [Author Response · Author response to Decision Letter 0]

12 Apr 2020

Response to editorial comments: 

Ok.

2. In ethics statement in the manuscript and in the online submission form, please provide additional information about the database used in your retrospective study. Specifically, please ensure that you have discussed whether all data were fully anonymized before you accessed them and/or whether the IRB or ethics committee waived the requirement for informed consent. If patients provided informed written consent to have their data used in research, please include this information.

Indeed, the IRB waived the requirements of informed consent. This information is included in the revised version of the manuscript.

"Maria Bogdanovskaya: I have read the journal's policy and the authors of this manuscript have the following competing interests: former employee of Dr Foster Telstra Health.

Lorna Barclay: I have read the journal's policy and the authors of this manuscript have the following competing interests: employee of Dr Foster Telstra Health."

We note that one or more of the authors are employed by a commercial company: Dr Foster Telstra Health.

“Competing interests related to the involvement of two authors need to be declared. MB and LB had an active role in the analysis of the data and participated in the design of the analysis of this manuscript. The company (Dr Foster Telstra Health) collected the data, but did not influence the study design and the data analysis. The company also did not have any influence on the decision to publish or the preparation of the manuscript. The involvement of two employees of the company does not alter our adherence to PLOS ONE policies on sharing data and materials. However, the data cannot be made available publicly as indicated below.” This statement is included in the competing interest section.

Data cannot be made publicly available as the terms and conditions of this collaboration restricted the availability of original data as follows:

1) Only certain categories of people are permitted to access and use the “Analytical Tool” (that is, the software processing input data and generating output data for providing benchmarking to each hospital for quality improvements). These are defined as:

“Permitted Users” means the Participant’s directors, employees and independent contractors who are authorised by the Participant to access and use the Analytical Tool for the purposes provided in this Agreement.

2) Access to and use of the Analytical Tool itself is granted only for the “Term” and for specific purposes, being “accessing and viewing the Output Data and to use the Output Data for the purpose of participating in and reviewing the results of the Project”. The restriction is related to the contract between the participating hospitals and Dr. Foster Telstra Health. There it was specified who has access to the data and that they may not be shared with other people as described above. Other researchers can request access to the data as long as they fulfil the requirements of the contract. Contracting the responsible person in each individual hospital is possible, but the authors cannot guarantee that each hospital will provide the data. This is the responsibility of each hospital. It would be all administrative data of the according years in the participating hospitals. The data are owned by the participating hospitals. A list of participating hospitals, [and their contact information] is included in the supporting information.

The following hospitals were participating in this study:

Hospital name City Country Contact Website

Alfred Health Melbourne Australia +61 3 90762000 www.alfredhealth.org.au

Austin Health Melbourne Australia +61 3 94965000 www.austin.org.au

Melbourne Health Melbourne Australia +61 3 93427000 www.thermh.org.au

Monash Health Melbourne Australia +61 3 95946666 monashhealth.org

Universitair ziekenhuis Leuven Leuven Belgium +32 16 332211 www.uzleuven.be/en

Aalborg UH Aalborg Denmark +45 97 666000 aalborguh.rn.dk/service/english

Guy's and St Thomas' London England +44 20 75895111 www.imperial.ac.uk

Imperial College Healthcare London England +44 20 33113311 www.imperial.nhs.uk

Royal United Hospitals Bath Bath England +44 1225 428331 www.ruh.nhs.uk

University College London Hospitals London England +44 20 34567890 www.uclh.nhs.uk

University Hospitals Coventry & Warwickshire Coventry England +44 24 76964000 www.uhcw.nhs.uk

Academisch Ziekenhuis Maastricht Maastricht The Netherlands +31 43 3876543 www.mumc.nl

Humanitas Research Hospital Milan Italy +39 02 82246250 www.humanitas.it

Barnes-Jewish Hospital St. Louis USA +1 314 7473000 www.barnesjewish.org

Hackensack University Medical Center Hackensack USA +1 844 4649355 www.hackensackumc.org

Hospital of the University of Pennsylvania Philadelphia USA +1 215 6624000 www.pennmedicine.org

Huntsville Hospital Huntsville USA +1 256 2651000 www.huntsvillehospital.org

Keck Hospital of USC Los Angeles USA +1 800 8722273 www.keckmedicine.org

UC San Diego Medical Center San Diego USA +1 858 6577000 health.ucsd.edu

University of Texas Southwestern Medical Center Dallas USA +1 214 6483111 www.utsouthwestern.edu

Yale–New Haven Hospital New Haven USA +1 203 6884242 www.ynhh.org

The according sentence (“Analysing only 2011-2013 did not affect results (data not shown).”) has been removed and is no longer part of the revised manuscript.

We have added the description of the link to the interactive figures 5A and 5B to the legend of figure 5 (previous figure 4), which now reads as follows:

“Figure 5 Link between original condition of admission (left side) and cause of readmission (right side) for each chronic condition. Figure B provides information about all other causes in figure A. These figures (Supporting information: Interacting Figures 4A and 4B) can be found online by clicking on the figures to provide a dynamic depiction of the different causes of readmission for each chronic condition”

In addition, we provided the following information at the end of the manuscript:

“Supplementary table 1: List of participating centres

Supplementary table 2: ICD 9 and ICD 10 codes for chronic conditions

Supplementary figures 5A and 5B: Link between original condition of admission (left side) and cause of readmission (right side) for each chronic condition. Figure B provides information about all other causes in figure A. Interactive online figures to provide a dynamic depiction of the different causes of readmission for each chronic condition”

Comments to comments by the reviewers

Reviewer #1:

The study is relevant, interesting and well written. I think this paper has the ability for a better understanding of the treatment and management of patients with chronic conditions and the described high rate of about 17% of readmissions. The conclusions could be strengthened. However, the present study is significantly limited by the administrative nature of the data, the lack of robust risk adjustment and the imbalances in the patient population.

We would like to thank the reviewer for carefully reading our manuscript and to provide positive feedback on it. 

We agree with the reviewer that the study has important limitations that we also addressed in part in the limitation section of the original submission. We now added the fact that risk adjustment was limited and that there may be an imbalance in the patient population. This is inevitable to administrative data. The advantage of administrative data is that there is no preselection of patients, which is inevitable to most registries. For the purpose of the study, we are convinced that administrative data are useful. 

There are some major concerns and limitations which have to be to addressed:

- The authors should clarify whether the identified ICD codes were main or secondary diagnoses for readmission of the patients. This fact could potentially bias the observed findings.

We agree with the reviewer. This is why we only used main diagnosis for readmission of patients to keep the potential bias as small as possible. We are aware that the coding may be subjective and not homogenous across all centres. This is addressed in the limitation section of the manuscript.

- Why did the authors choose the 30 days as time point of emergency readmission?

To some extent, this is obviously arbitrary. However, the reason for this is mainly twofold: a) reimbursement policy in the USA is related to the 30-day readmission rate. The result of this fact is that 30-day readmission is often used in publications, which makes comparison more accurate; b) we wanted to avoid interference with new diseases that develop over time to really be able to focus on readmissions related to the initial diagnosis. We did not include this in the revised manuscript as we do not think that this is a crucial part for the understanding. However, if the editor feels that we should add a short paragraph to highlight these thoughts, we are certainly willing to do so.

- The authors should stratify their findings concerning AMI as reason of readmission in the subtypes of AMI and by OPS codes for coronary interventions. These findings should also be incorporated in the rates of in-hospital mortality and might show differences over the time and by area (country).

Unfortunately, this information is not available with sufficient accuracy. Due to countries having different coding versions, the coding is not sufficient across all countries and sites for this analysis. This is an important reason why we did not split diagnoses / readmissions in further details. Moreover, it would not change the frequency of readmissions due to causes other than the index event. Thus, the main message of the study is not influenced by this. Still, we agree with the reviewer that this would be interesting in a study that specifically addresses the issues of readmissions in AMI patients. However, this is beyond the scope of this study.

- Table 4 shows the calculated odds-ratios of regression for probability of readmission with the same condition as initial hospitalisation. The authors should adjust the findings i.e. by area (country), length of stay and year of admission.

We agree with the reviewer that this is interesting, which is why we did the multivariable regression analysis as depicted in table 4. Thus, the odds ratios provided are adjusted by all the variables included. Obviously, this was not clear which is why we added that the table contains the results of multivariable regression.

Reviewer #2:

The issue of readmission rate in chronic disease is of outmost social and economic importance, but it has been often regarded from a disease-specific point of view. The Authors should be commended for their attempt to provide a picture of the possible paths patients with major chronic conditions often follow, using a huge amount of administrative data from all over the world.

We would like to thank the reviewer for his/her positive comment on our study.

While of interest, the value of the findings of the paper is limited from some major methodological issues, as reported below in details.

Major points

- The Authors state that they “included two acute conditions, i.e. AMI and bacterial pneumonia”. Indeed, stroke (acute?!) was also considered in the analysis. Although they were apparently included as they underly other chronic conditions, the Reviewer is rather concerned about this choice.

We agree with the reviewer that stroke is also an acute event, based on an underlying chronic disease. We changed this accordingly in the revised manuscript.

Moreover, we understand the concern by the reviewer. In fact, we had some discussion about this point when designing the analysis. We decided to do so as we aimed to provide a broad picture of readmission of important different chronic diseases and that the underlying disease is most important for them. Also the other included conditions usually have acute events that lead to hospitalisation (e.g. acute decompensation of heart failure or CODP, occurrence of atrial fibrillation). In fact, the readmission rate of these three acute conditions were in line with the other conditions, supporting our choice. Therefore, we did not change the diseases included in the analysis. Nevertheless, we included the following paragraph to the limitation section:

“We included conditions that present with acute events (i.e. AMI, pneumonia, stroke) but are usually caused by a chronic underlying disease. Including these conditions might have influenced our results. However, readmission pattern of them were in line with the other more chronic conditions, supporting the assumption that the readmissions were mainly influenced by the underlying chronic condition rather than the acute event.”

- Only readmissions at the same hospital could be recorded. Although the disease specific readmission rates are similar to those reported in the past (for example 7 vs 5% for heart failure, Fudim M et al. Aetiology, timing and clinical predictors of early vs. late readmission following index hospitalization for acute heart failure: insights from ASCEND-HF. Eur J Heart Fail. 2018;20:304-314), the whole number of readmission may have been significantly underestimated, especially as concerns those caused by different diseases (as patients may have been referred to different specialized hospitals).

We fully agree with the reviewer which is why we included this discussion point in the original limitation section. In addition, we added the comparison with heart failure as an example as suggested by the reviewer.

- Administrative and reimbursement issues may have significantly influenced the attribution of the readmission cause, thus possibly biasing the Authors’ findings.

We fully agree with the reviewer. This is why we mention this point as our first limitation to our study. As also highlighted in one of the points raised by reviewer #1, there are pros and cons for the use of administrative data for this kind of analyses. For the purpose of our study, we are convinced that the inclusion of all cases of a specific hospital is an important pro, allowing the conclusions made.

- How was time to first readmission related to the readmission cause?

This is an interesting point. In fact, the time to readmission was exactly the same if comparing similar or different cause of readmission overall. There were small differences between the different original conditions. Due to the large number of patients, they obviously are all statistically significantly different, but of clinical relevance, only two are meaningful. We added this information to the manuscript in a separate paragraph of the Results section and added a figure (new figure 4).

Minor points

- How do the Authors explain their observation that readmission rated have not changed significantly along the whole study period for most of the chronic conditions they have considered?

We tend to disagree with the conclusion by the reviewer that there were no changes over time. In fact, there was a significant reduction in admission due to the same condition over time (1.3% per year) as indicated in table 4. As discussed, we hypothesise that this is related to the increase in age / complexity of chronic diseases over time.

- Discussion should be shortened and focused on the main findings of the study.

We agree in part with the reviewer and have shorten the discussion. We are also aware of the fact that our data are not sufficiently in depth to address all questions related to the topic. In order to stimulate the discussion and additional research in the field – both to better understand and to prevent readmissions – we deliberately discussed some issues where our results are more hypothesis generating than providing clear evidence. 

Reviewer #3: 

Thank you for the opportunity to review this manuscript. It explored causes of admission and readmission in a unique population of multiple academic centers across the world. In a type of consortium 20+ centers have pooled administrative data for purposes of quality improvement research. IN this analysis the researchers found that in most cases the cause of readmission is different from the cause of admission the first time around. These findings have been previously described in a variable type of cohorts including administrative databases and clinical trial cohorts (see my last comment). Nevertheless this data remains very important and only underlines the need for a more “holistic” patient care (as the authors called it).

We would like to thank the reviewer for the positive comment on our study.

Comments:

- This study encompasses multiple countries. US policy on readmission reduction was cited but what about other countries? Do they have comparable initiatives?

Although each healthcare system aims to reduce readmission rates, the US policy in this regard is rather unique. This is why we focused on discussing this in our manuscript. Given the fact that reduction in the length of the discussion is suggested by this and another reviewer, we have not extended our discussion in this point. 

- I am not sure I follow the difference between ICD and CCS. Aka Table 1 and 2.

The ICD code defines diseases in more detail, where as the CCS summarises diseases in groups. It is referenced in the paper (see Methods section, ref 17)

- It is important to understand where the ICD code for the primary diagnosis was derived from. Did the ICD code get obtained from the billing information, discharge notes etc? Only the code I the first position was used?

Administrative dataset contain data that are used for national registers and are obtained from discharge notes. As mentioned in the manuscript, only the main diagnosis was used.

- Found the amount of information provide in the figures to be somewhat overwhelming.

We understand this notion by the reviewer. However, we are convinced that this is a strength of the manuscript as information is available if readers are interested in the details of our findings. This is why we did not reduce it.

- Abbreviations in all tables and figures need to be presented in legends.

We agree with the reviewer and changed this accordingly.

- Information in the discussion section is repetitive. Reduce repetition and reduce in length.

We agree in part with the reviewer and have shorten the discussion, particularly to avoid repetition unless required to understand the context of the according paragraph. We are also aware of the fact that our data are not sufficiently in depth to address all questions related to the topic. In order to stimulate the discussion and additional research in the field – both to better understand and to prevent readmissions – we deliberately discussed some issues where our results are more hypothesis generating than providing clear evidence. 

- Given that HF is a major contributor to the admission and readmission consider to discuss some of the already published literature on this topic. PMID: 27133201 and PMID: 29082629.

We agree with the reviewer that HF is an important chronic condition. As also suggested by one of the other reviewers, we added one of the two suggested references to the manuscript, mentioning HF as example for comparison with our results.

---

## [Decision Letter · Decision Letter 1]

6 May 2020

Reasons for readmission after hospital discharge in patients with chronic diseases  - information from an international dataset

PONE-D-19-35626R1

Dear Dr. Brunner-La Rocca,

We are pleased to inform you that your manuscript has been judged scientifically suitable for publication and will be formally accepted for publication once it complies with all outstanding technical requirements.

With kind regards,

Gianluigi Savarese

Academic Editor

PLOS ONE

Additional Editor Comments (optional):

Reviewers' comments:

Reviewer's Responses to Questions

**Comments to the Author**

1. If the authors have adequately addressed your comments raised in a previous round of review and you feel that this manuscript is now acceptable for publication, you may indicate that here to bypass the “Comments to the Author” section, enter your conflict of interest statement in the “Confidential to Editor” section, and submit your "Accept" recommendation.

Reviewer #1: All comments have been addressed

Reviewer #2: All comments have been addressed

Reviewer #3: All comments have been addressed

2. Is the manuscript technically sound, and do the data support the conclusions?

Reviewer #1: Yes

Reviewer #2: Yes

Reviewer #3: Yes

3. Has the statistical analysis been performed appropriately and rigorously? 

Reviewer #1: Yes

Reviewer #2: Yes

Reviewer #3: Yes

4. Have the authors made all data underlying the findings in their manuscript fully available?

Reviewer #1: Yes

Reviewer #2: Yes

Reviewer #3: Yes

5. Is the manuscript presented in an intelligible fashion and written in standard English?

Reviewer #1: Yes

Reviewer #2: Yes

Reviewer #3: Yes

6. Review Comments to the Author

Reviewer #1: Dear Editor,

the authors were able to answer all queries extensively and sufficiently.

In my opinion the present study explored causes of admission and readmission in a unique population.

From the reviewer´s view, the manuscript can be accepted for publication in PLOS ONE.

With best regards and thanks in advance,

Moritz Becher

Reviewer #2: The Authors have addressed, as far as possible, the issues raised in the previous revision of the manuscript.

Reviewer #3: Thank you for addressing my comments. I have no further suggestions at this time.

..................

7. PLOS authors have the option to publish the peer review history of their article (what does this mean?). If published, this will include your full peer review and any attached files.

Reviewer #1: No

Reviewer #2: No

Reviewer #3: Yes: Marat Fudim

---

## [Editor Report · Acceptance letter]

20 May 2020

PONE-D-19-35626R1 

Reasons for readmission after hospital discharge in patients with chronic diseases  - information from an international dataset 

Dear Dr. Brunner-La Rocca:

I am pleased to inform you that your manuscript has been deemed suitable for publication in PLOS ONE. Congratulations! Your manuscript is now with our production department. 

With kind regards,

on behalf of

Dr. Gianluigi Savarese 

Academic Editor

PLOS ONE